# Effect of Concentration of Flaxseed (*Linum usitatissimum*) and Duration of Administration on Fatty Acid Profile, and Oxidative Stability of Pork Meat

**DOI:** 10.3390/ani12091087

**Published:** 2022-04-22

**Authors:** Martin Bartkovský, Drahomíra Sopková, Zuzana Andrejčáková, Radoslava Vlčková, Boris Semjon, Slavomír Marcinčák, Lukáš Bujňák, Matej Pospiech, Jozef Nagy, Peter Popelka, Petronela Kyzeková

**Affiliations:** 1Department of Food Hygiene, Technology and Safety, University of Veterinary Medicine and Pharmacy in Košice, Komenského 73, 041 81 Košice, Slovakia; boris.semjon@uvlf.sk (B.S.); slavomir.marcincak@uvlf.sk (S.M.); jozef.nagy@uvlf.sk (J.N.); peter.popelka@uvlf.sk (P.P.); 2Department of Biology and Physiology, University of Veterinary Medicine and Pharmacy in Košice, Komenského 73, 041 81 Košice, Slovakia; zuzana.andrejcakova@uvlf.sk (Z.A.); radoslava.vlckova@uvlf.sk (R.V.); 3Department of Animal Nutrition and Husbandry, University of Veterinary Medicine and Pharmacy in Košice, Komenského 73, 041 81 Košice, Slovakia; lukas.bujnak@uvlf.sk; 4Faculty of Veterinary Hygiene and Ecology, University of Veterinary Sciences Brno, Palackého Tr. 1946/1, 61242 Brno, Czech Republic; pospiechm@vfu.cz; 5Clinic of Swine, University of Veterinary Medicine and Pharmacy in Košice, Komenského 73, 041 81 Košice, Slovakia; petronela.kyzekova@uvlf.sk

**Keywords:** pigs, meat quality, fatty acid composition, fat

## Abstract

**Simple Summary:**

Dietary fat intake is substantially high in Western countries, resulting in overweight, obesity and cardiovascular diseases (CVD) among consumers. One way to reduce the incidence of CVD is to change food consumption and food intake with a higher content of polyunsaturated fatty acids (PUFA) in the human diet. Meat and meat products are considered to be the main source of dietary fats, especially as a source of saturated fatty acids. The aim of this study was to increase the proportion of PUFAs and, conversely, to decrease the proportion of saturated fatty acids in fat of pork meat. Flaxseed with a concentration of alpha-linolenic acid (ALA) at 57% was used in the swine diet. Flaxseed supplementation at two inclusion levels (5% and 10%) in two time intervals (3 and 6 weeks before slaughter) was evaluated. The aim was to increase the proportion of *n*-3 PUFAs, especially ALA, eicosapentaenoic acid (EPA), docosapentaenoic acid (DPA) and docosahexaenoic acid (DHA) and to improve the *n*-6/*n*-3 PUFA ratio in produced pork meat. The storage conditions and oxidation stability of the produced meat under refrigerator temperature (+4 °C) were also observed.

**Abstract:**

Flaxseed is a common ingredient used for livestock feed. The aim of this work was to study the effect of a diet supplemented with flaxseed at 5% and 10% concentrations in the intervals of 3 and 6 weeks prior slaughter on fatty acid profile and oxidative stability of pork meat. Meat samples were collected after slaughter from each animal (five groups, *n* = 6). Samples of the *musculus longissimus dorsi* (MLD) and the *musculus gluteobiceps* (MGB) were selected. Chemical composition, fatty acid profile and oxidative stability during the storage of meat under chilling conditions (4 °C, 7 days) was analyzed. The addition of flaxseed significantly affected the composition of fatty acid profile and the shelf life of the produced meat. The fat content was changed in the experimental groups with 10% flaxseed supplementation (10.84% in MGB and 9.56% MLD) versus the control group. Despite the different concentrations of flaxseed, the best EPA/AA ratio was observed in the experimental groups fed with flaxseed supplementation for 3 weeks. The worst oxidative stability of meat samples (*p* < 0.05) was recorded in the experimental groups with the addition of flaxseed for 6 weeks, which was related to higher PUFA content in samples of the experimental groups and higher susceptibility of PUFAs to lipid oxidation. The oxidative stability of meat in the experimental group fed 5% flaxseed supplementation for 3 weeks was not affected.

## 1. Introduction

The diet structure of most consumers in Europe does not meet the recommendations of doctors and dieticians [1]. Polyunsaturated fatty acids (PUFAs) of *n*-3 and *n*-6 characteristics do not have the same metabolic rate and the organism does not use the same enzyme complex for their metabolism. The importance of PUFAs in diet is not only in their sufficient income, but also in the ratio of the *n*-3/*n*-6 PUFAs. Both linolenic acid (LA; 18:2 *n*-6) of *n*-6 line and alfa-linolenic acid (ALA; 18:3 *n*-3) of *n*-3 line belong to the essential PUFAs. Each have importance and act as precursors to many biologically active substances [2,3]. Deficiency of these PUFAs significantly affects the level of lipid composition in animal tissues and membranes of their cell structures [3]. Despite all the recommendations of the WHO, the human population does not receive sufficient amounts of essential PUFAs [4]. Daily intake of eicosapentaenoic acid (EPA, 20:5 *n*-3) and docosahexaenoic acid (DHA, 22:6 *n*-3) is well below the limit in Western Europe and North America [2,4]. Fat intake in Western countries (USA, EU, etc.) is mainly in the form of vegetable oils and animal fats with a higher content of LA and ALA in produced food, where the nutritional trends fail to provide enough EPA and DHA [2]. The addition of fat and oil supplements with a higher proportion of PUFAs to animal feed is an effective method for increasing the content of PUFAs in the meat of slaughter animals. Several lipid sources have been tested for pig feeding with the aim to improve the *n*-3/*n*-6 ratio of meat. In recent years, most attempts have focused on the application of vegetable oils, especially linseed oil [4], rapeseed oil [5] and olive oil [6,7]. 

In monogastric animals, it is possible to enhance the concentration of beneficial *n*-3 fatty acids in the muscle tissue since meat lipids reflect the nature of the dietary fat. Feeding whole flaxseed to pigs increases the *n*-3 fatty acids in various tissues without an effect on growth and meat quality during storage [8,9]. Flaxseed could be fed in various forms and doses. Lower doses (up to 5%) were examined by Matthews et al. [8], Riley et al. [9], Kouba et al. [10] and others. Juárz et al. [11] reported that higher doses of flaxseed supplementation, up to 20%, had no effect on growth performance. Flaxseed diet, which is rich in ALA, could cause an increase in *n*-3 PUFA deposition in tissues and decrease the atherogenic and thrombogenic index of pig meat [12,13]. The possibility to incorporate natural vegetable-based antioxidants in the pig diet to enhance the nutritional and health benefits of meat represents a new approach in the supply chain of meat production [14]. Simultaneously, the administration of PUFA-rich ingredients (such as flaxseed) in the pig diet could delay the oxidation of meat lipids and also enrich meat with selected fatty acids [15]. 

This experiment was focused on improving the quality of pork meat supplementing pigs’ diet with flaxseed (*Linum usitatissimum*). We investigated the effects of supplementation of two different concentrations of flaxseed at various intervals during two fattening periods on the quality of produced pork meat. The changes in the FA profile and the ratio of *n*-3/*n*-6 PUFAs in pork meat were investigated. The lipid oxidation processes in meat samples were evaluated via the malondialdehyde (MDA) content, a main product of PUFA oxidation during the meat storage period. 

## 2. Materials and Methods

### 2.1. Animals, Diets and Management

The experiment was carried out in accordance with the European Directive (Directive 2010/63/EU, 2010) on the protection of vertebrate animals used for experimental and other scientific purposes (2010/63/EU).

A total of 30 Landrace breed pigs (gilts) were used in the experiment, which lasted 6 weeks. Pigs used in this experiment were 20 weeks old at the beginning of this experiment, with live weight of 76.61 ± 2.30 kg. Animals were kept in pens (groups of 2 animals per pen; 4.3 m^2^) equipped with nipple drinkers. Average temperature was 18 °C and humidity was 60%. Health status, consistency of feces and body weight of pigs were monitored daily, after 21 days of supplementation and at the end of experiment. Animals underwent parasitological and bacteriological observations of feces samples and nasal swabs with negative results. On the day of slaughter, the animals weighed about 118.16 ± 4.84 kg. Pigs were divided into 5 groups. During flaxseed supplementation, animals were weighed three times (Table 1. The control group (C) was fed a commercial complete fattening feed. The first experimental group was fed the commercial complete feed mixture supplemented with flaxseed in a 5% dose for 3 weeks (F5W3). The second experimental group was fed a 5% concentration of flaxseed for 6 weeks of fattening (F5W6). The third experimental group was fed the commercial complete feed mixture supplemented with 10% flaxseed for 3 weeks (F10W3). The fourth experimental group was fed a 10% concentration of flaxseed for 6 weeks (F10W6). 

Gilts were housed at the Pig Fattening and Slaughter Station Inc. (Vajanského street 789, Spišské Vlachy, the Slovak Republic). The commercial complete feed mixture for fattening pigs was used (Dom krmív, Spišské Vlachy, the Slovak Republic) at a dose of 3 kg/day. The main ingredients of the commercial feed were maize (13%), alfalfa (36.5%), wheat (32%), soybean meal (8.5%) and wheat bran (7%). The diet also contained bicalcium phosphate (1.29%), monocalcium phosphate (0.78%), sodium chloride (0.45%) and mineral premix (0.48%). During the experimental period, the pigs’ diet was supplemented with flaxseed (Libra variety, ALA acid content 57%). The chemical compositions of pigs’ diets are presented in Table 2 and Table 3. The animals had free access to water. 

### 2.2. Meat Sample Preparation

Loin (*musculus longissimus dorsi*) and thigh meat (*musculus gluteobiceps*) from each carcass were taken for the meat quality evaluation. The central part of the muscle (loin and thigh meat) with no adhering subcutaneous or intermuscular adipose tissue was examined. Whole pieces of *m. longissimus dorsi* and *m. gluteobiceps* were removed. Muscle pieces were subsequently divided into equal anatomical parts. The total weight of each sample was set at 1000 g. Samples were divided according to experimental groups depending on the flaxseed concentration and on the interval of fattening. Meat samples were packed under vacuum conditions and stored at a temperature of 4 ± 2 °C until analysis.

### 2.3. Determination of Meat Quality Parameters

The content of dry matter in meat samples was determined by oven-drying at a temperature of 105 ± 2 °C [16]. Kjeltec Auto, type 1030 analyzer (Tecator Co., Hoganas, Sweden) was used to determine the crude protein content according to the Kjeldahl method. Lipid content was measured in ground samples with petroleum ether by the gravimetrical method in Soxhlet apparatus (LTHS 500, Brnenská Druteva v.d., Brno, Czech Republic).

The FA extraction of loin and thigh meat was carried out according to the Folch method [17]. Extracted lipids were transesterified to fatty acid methyl esters (FAME) with sodium methanolate. The profile of fatty acid methyl esters was analyzed by a gas chromatograph with a flame ionization detector (Shimadzu GC 17, Shimadzu, Japan). The results of FA profiles were expressed as percentages of the total fatty acids calculated as a mean value of three measurements.

The extent of lipid oxidation in loin and thigh meat samples was evaluated as thiobarbituric acid reactive substances (TBARS) according to Reitznerová et al. [18]. TBARS values were measured spectrophotometrically at 532 nm (Helios α, v. 4. 6. Thermospectronic, Cambridge, UK). TBARS values were determined within 24 h after slaughter, and each sample was analyzed within the next five and seven days of storage in a refrigerator at a temperature of 4 ± 2 °C. The results were quantified as malondialdehyde (MDA) equivalents and expressed as mg of malondialdehyde/kg of sample.

### 2.4. Statistical Analyses

Data analysis was carried out via Graph Pad Prism 8.3 (GraphPad Software, San Diego, CA, USA). The results of each variable are expressed as mean and standard error of the mean (SEM). One-way analysis of variance (ANOVA) was used to evaluate statistical significance using Tukey’s multiple comparisons between control and experimental groups, with a significance level set at *p* < 0.05.

## 3. Results

Table 2 shows the nutritional composition of pigs’ diets calculated as 100% of dry matter. After the 5% and 10% supplementation of flaxseed, an increase in the proportion of protein and fat in the experimental diets was observed. The metabolizable energy (ME) of the diets was also improved with both concentrations of flaxseed supplementation (5% and 10%). Although the starch content was lower in the experimental diets, the metabolizable energy balance was not influenced.

The results of meat composition determinations are shown in Table 4. A statistically significant increase in the proportion of dry matter of *m. gluteobiceps* (*p* < 0.05) was found.

Significantly higher dry matter variation was observed in the meat samples of experimental groups fed with a diet supplemented with flaxseed for 3 weeks (F5W3 and F10W3) compared to groups in which flaxseed was administered for 6 weeks, as well as compared with the control.

The highest fat content was recorded in F10W3, and in the F5W3 experimental group, the highest protein content of 21.95 ± 0.23 was observed in *m. gluteobiceps*. Analyzing the *m. longissimus dorsi* samples, statistically higher proportions of dry matter and fat content were recorded in pigs fed with flaxseed for 3 weeks (F5W3 and F10W3), when compared to the control group (*p* < 0.05). Flaxseed supplementation for 6 weeks did not affect the dry matter and fat content for the *m. longissimis dorsi* samples (*p* > 0.05) from middle part of the muscle. The results of these chemical variables were comparable to the control group. Protein content of meat samples was not affected (*p* > 0.05). 

The lipid oxidation and storability of the fat samples are directly dependent on the proportion of FA. The TBARS method was used to determine the lipid oxidation products (MDA content). The results of lipid oxidation of samples during storage under chilling conditions (4 °C, 7 days) are presented in Table 5. The supplementation of flaxseed directly affected the composition of FA and increased the susceptibility of the meat fat component to oxidation. We observed an increase (*p* < 0.05) in the amount of MDA in F10W6 samples (0.276 mg.kg^−1^) of *m. gluteobiceps*. On the seventh day of meat storage, the highest amount of MDA in the F5W6 sample was recorded. Fats from pigs fed flaxseed for 6 weeks are more susceptible to lipid oxidation during storage. The oxidative stability of F5W3 samples was not affected and was comparable with the control. Samples of *m. longissimus dorsi* showed the highest MDA values in the F10W6 samples on the seventh day of storage (0.325 mg.kg^−1^), followed by F5W6 samples (0.302 mg.kg^−1^), and compared to the control, oxidation damage in F5W3 samples was not higher.

The fatty acid profiles of *m. gluteobiceps* meat samples are presented in Table 6. The contents of PUFAs in *m. gluteobiceps* samples were statistically influenced by concentration and time of exposure of flaxseed feeding (*p* < 0.05). Higher concentrations of ALA in all experimental groups were found. The concentration of LA was significantly lower in experimental groups compared to the control (*p* < 0.05). The highest value of ALA was observed in the F10W6 group. The lowest value of LA was observed in the F10W3 group. Six-week supplementation of flaxseed significantly increased the content of EPA, DPA and DHA in samples of F5W6 and F10W6 experimental groups compared to the control. Supplementation of flaxseed for 3 weeks did not affect the EPA, DPA or DHA content, and the concentrations of these PUFAs were comparable to the control. The *n*-6/*n*-3 ratio was significantly lower in all experimental groups compared to the control (*p* < 0.05). The lowest *n*-6/*n*-3 ratio was recorded in the F10W6 experimental group.

The fatty acid profile of *m. longissimus dorsi* is presented in Table 7. Flaxseed addition changed the composition of selected PUFAs in the meat of experimental groups. Higher concentrations of ALA, ARA, EPA, DPA and DHA in all experimental groups were recorded. Compared to the control group, LA concentrations were significantly lower (*p* < 0.05) in all experimental groups. The highest concentrations of ALA and DPA were found in the F10W6 experimental group, and concentrations of ARA, EPA and DHA were higher in the F5W6 experimental group. The effect of the length of flaxseed supplementation was evident in the profile of FA of experimental groups. Higher content of the mentioned PUFAs in the samples after 6 weeks of flaxseed supplementation was recorded in comparison with the 3-week supplementation interval. Supplementation with 5% flaxseed had a greater effect on concentrations of ARA, EPA and DHA after 6 weeks of supplementation. The effect of 3 weeks of supplementation of different concentrations was significant, as the contents of all *n*-3 PUFAs were higher in the F10W3 experimental group compared to F5W3. The *n*-6/*n*-3 ratio was significantly lower in all experimental groups compared to the control (*p* < 0.05). The lowest *n*-6/*n*-3 ratio was observed in the F10W6 experimental group.

## 4. Discussion

In the past few years, several attempts have been made to modify the FA composition of the carcass lipid depots through feeding strategies to meet human dietary recommendations. In recent years, according to specific dietary guidelines for fat by health institutions worldwide, it is encouraged to limit the consumption of total fat in a healthy dietary pattern [19]. Low PUFA/SFA and high *n*-3/*n*-6 PUFA ratios of red meat (pork, beef, etc.) are well known to contribute to the fatty acid intake imbalance in consumers [20]. These nutritional recommendations for FA have raised substantial interest because of the impact of total fat and specifically for different FAs associated with different health effects and nutritional well-being [15]. PUFAs have an important role in antithrombotic and anti-inflammatory processes in the body [3,15]. Linolenic acid is derived entirely from the diet. It passes through the pig’s stomach unchanged and is then absorbed into the blood stream in the small intestine and incorporated into tissues. The second most important PUFA is ALA, which is presented in many concentrate feed ingredients but at lower levels than LA. In pigs, the proportion of this FA is higher in adipose tissue than muscle [21]. The ability to synthesize the appropriate FA is affected by the amount of desaturase activity, age, infections and nutritional factors. Vitamin deficiency indeed affects desaturase and elongase, but it is not relevant to the current trial [3]. Polyunsaturated fatty acids from *n*-3 and *n*-6 lines have the same metabolic rate and the organism uses the same enzyme complex for their incorporation to the fat tissue. Higher PUFAs and a lower *n*-6/*n*-3 ratio will positively affect the fat deposition in pigs [22]. The addition of supplements, fats and oils with a higher proportion of PUFAs in the feed is an effective method for increasing the content of PUFAs in the fat of slaughter animals. Most attempts were focused on the addition of vegetable oils [7]. Kim et al. [23] presented a study on using 50.0 g/kg linseed oil to finishing pigs’ diets. Concentrations of adipose tissue content of *n*-3 and *n*-6 were changed, whereas concentrations of SFA were reduced. There are some limitations to the modification of the fatty acid profile in pork meat, as increased content of PUFAs can negatively influence nutritional and sensory parameters [24]. Rey et al. [25] also proved growth of PUFAs after linseed oil consumption.

Products including arachidonic acid (20:4*n*-6, ARA) and eicosapentaenoic acid (20:5*n*-3, EPA) have various metabolic roles including eicosanoid production. Dietary lipid supplementation with vegetable oils has shown to increase the amount of carcass backfat in finisher pigs [22,26]. The same results were observed in our research. There are also a lot of inconsistent findings among studies of produced backfat, which can be explained by metabolizable energy content in control diets [27] and energy status of the experimental animals [28], but also genetic factors [29]. Many studies found no effect of dietary lipid supplementation on the energy ratio of the feed mixtures [30,31]. The effect of lipid addition to commercial feed may have contributed to the inconsistency of these reports, and dietary lipids can reduce de novo lipogenesis [32]. While SFA and MUFA are de novo synthesized and their concentrations are less influenced by diet, the essential PUFAs cannot be synthesized in situ and have to be incorporated directly into tissue with concentrations more predisposed to dietary changes. Typically, oleic acid (OA, 18:1c9) is the major fatty acid in porcine meat and it is more predominant in neutral lipids, whereas LA and ALA are mostly found in membrane phospholipids than in triacylglycerols. The higher incorporation of LA into pig muscle yields superior proportions of AA and higher *n*-3/*n*-6 PUFA ratios compared to ruminants [33]. The determination of metabolizable energy (ME) is shown in Table 2. ME values in our experimental diets with 5% and 10% flaxseed supplementation were 13.31 MJ/kg and 13.36 MJ/kg, respectively, both of which were higher than in the study of Huang et al. [32]. The measured ME values are similar to canola meal values presented in the study of Liu et al. [34] and lower than doses reported for soybean [35] and camelina meal [36]. The content and composition of the body fat is the result of lipid anabolism and catabolism in growing pigs. The de novo lipogenesis accounts for at least 74% of adipose tissue triacylglycerols (TAG) [37] using corn- and soybean-based diets, which usually contain up to 4% of lipids [38,39,40]. Juárez et al. [41] examined the effect of gender on the backfat FA composition. Different feed time intervals of flaxseed proved the effect on the FA composition. Similar results were reported by Nuernberg et al. [42] and likely relate to barrows being fatter with lipids being relatively rich in SFA and *n*-6 PUFA [9]. All differences between genders were, however, less than 1% of total FA and would be of limited practical significance. Lower tissue levels of docosahexaenoic (DHA, 22:6*n*-3) acid were reported by several authors [9,43,44] after feeding pigs dietary flaxseed in doses up to 2.5%. Relatively few studies reported increased levels of DHA after flaxseed addition [45,46]. Long-chain fatty acid metabolism is controlled by complex enzymatic systems [43]. Enzymes such as desaturases and elongases act on the *n*-3 and *n*-6 fatty acids but have preference to the *n*-3 [47]. Diet, feeding length and amount of flaxseed (*p* < 0.05) affect the concentrations of total PUFAs, monosaturated fatty acids (MUFAs) and SFA. The *n-3/n-6* ratio was influenced by flaxseed (Table 6 and Table 7). Diets containing flaxseed changed 18:3*n*-6, 18:3*n*-3 and 20:5*n*-3 amounts in experimental diets. The best results for PUFA profile were recorded in groups with 5% concentrations of flaxseed. The ARA content was lower in experimental diets. During lipogenesis, the main precursor for the production of ARA is LA. If there is a deficiency in LA content higher amounts, of ARA occur. We observed statistically significant (*p* < 0.05) differences in LA and ARA content in experimental groups, and there was also a difference between *m. longissimus dorsi* and *m. gluteobiceps* in LA and ARA content. Adding flaxseed to pig diets does not only affect the levels of *n*-3 FA; there was also a decrease in relative content of 18:2*n*-6 (linolenic acid, LA). The decrease in 20:4*n*-6 levels in *m. gluteobiceps* experimental groups may be due to competition between 18:2*n*-6 and 18:3*n*-3 for desaturation and elongation to form 20:4*n*-6 and 20:5*n*-3 [48]. The decrease in 20:4*n*-6 levels might be beneficial, since the increase in this FA in membranes of phospholipids results in an overproduction of eicosanoids [49]. Animals fed with 10% flaxseed had lower content of 18:3*n*-6 compared to control group. FA composition of *m. gluteobiceps* was almost the same as the *m. longissimus dorsi* samples. The highest concentration of α-linolenic acid as a precursor for *n*-3 fatty acids was observed in the sample of *m. longissimus dorsi* (9.507%), which was higher compared to *m. gluteobiceps* fat (7.160%). Compared to Juárez et al. [41], our flaxseed concentrations (5%, 10%) were lower, but we achieved similar results for fatty acid profile change. The present study confirmed findings previously published by Guillevic et al. [50] and Okrouhlá et al. [12], who showed that the main effect of dietary linseed addition is reflected in the fatty acid profile of the pork meat, when palmitic and stearic acids are the dominant acids among the saturated fatty acids (SFAs) [12]. Higher contents of LA and ALA in pork meat were observed in the study of Bečková and Václavková [51]. Flaxseed was added to the feed mixture, and a negative effect on arachidonic acid was observed. Flaxseed statistically changed the content of LA, ALA, EPA and AA. LA is a precursor for *n*-6 PUFA synthesis, and according to our results, 3 weeks of flaxseed application to the pigs’ diet was not sufficient to incorporate this FA into meat samples. These observations were confirmed by higher amounts of GLA in experimental diets compared to the control group. The balanced amount of ALA resulted in an increase in EPA and DHA concentrations in produced meat samples. The experimental groups showed better *n*-6/*n*-3 ratios than the control group (Table 7). According to Enser et al. [45], flaxseed feeding increased essential fatty acid content. Okrouhlá et al. [12] observed effects of flaxseed on the PUFAs comparable with our results. The best results of the *n*-3/*n*-6 ratio were observed in the F10W6 group in *m. longissimus dorsi* and *m. gluteobiceps*. The *n*-3/*n*-6 ratio in *m. longissimus dorsi* was significantly (*p* < 0.05) influenced by dietary flaxseed. With a different flaxseed time exposure and concentration, we achieved a high proportion of *n*-3 PUFAs. Higher concentration and longer time of exposure increases the proportion of LA and ALA as precursors for the *n*-3 and *n*-6 FA groups. Statistically significant (*p* < 0.05) ALA concentrations were seen after 6 weeks at a 10% flaxseed dose. α-linolenic acid as a precursor for *n*-3 FA caused EPA and docosapentaenoic (DPA) acid increase. Feeding pigs higher concentrations of flaxseed for shorter periods versus lower concentrations for longer periods appears to be more efficient at increasing *n*-3 PUFAs [41]. Different studies compare the amounts of flaxseed in the feed for *n*-3 PUFA ratio changes in pork meat [10,41,52]. Juárez et al. [41] investigated overall changes in backfat levels of *n*-3 and *n*-6 PUFAs after feeding with 10% and 15% co-extruded flaxseed. Experimental feeding showed a linear increase in PUFAs; however, mean concomitant reductions in SFA and MUFA were observed [41]. In the present study, we observed increased content of *n*-3 PUFAs—ALA, EPA and DHA—in correlation with the supplementation of flaxseed in the pigs’ diet. According to our results, it can be concluded that a higher dose of flaxseed supplementation for a short period was beneficial in terms of increasing the content of *n*-3 PUFAs, as mentioned above. On the other hand, a lower dose of flaxseed in pigs’ diet for a longer period had a similar effect. It can be considered that the optimal administration of flaxseed to pigs is 5% flaxseed supplementation for 6 weeks.

The beneficial effects of *n*-3 PUFAs, especially of EPA and DHA, in the prevention of chronic diseases have been well recognized over the past few decades [53,54]. Oxidation directly affects sensory properties, which are significantly worsened during storage and heat treatment. Muscle contains significant proportions of long-chain PUFAs, which are formed from 18:2*n*-6 and 18:3*n*-3 by the desaturase and elongase enzymes action. In fact, PUFAs are prone to oxidation due to the presence of unsaturated bonds, leading to the production of different metabolites, which may adversely affect human health, shelf life and meat quality [33]. Susceptibility of PUFAs to oxidation depends on multiple factors, such as the presence of antioxidants, pro-oxidants, storage conditions, etc. Lipid peroxidation is the most aggressive chemical reaction in foodstuff [17,55]. Malondialdehyde (MDA) is the main secondary lipid oxidation product [56]. Compared to the other secondary products of lipid peroxidation, MDA is stable and abundant, considered as one of the most important markers of lipid peroxidation status [57,58]. Due to the presence of carbonyl groups, MDA is an easily reactive molecule that undergoes several reactions with nucleic acids, proteins and lipoproteins, changing their chemical behavior [58,59]. The presence of MDA has been reported in various biological matrices, such as serum, urine, etc., and moreover, in animal and vegetable foodstuff. Determination of MDA seems to be an appropriate food quality marker [55,58,60]. 

The addition of flaxseed increased MDA values in the experimental groups, as expected (Table 5). The highest values (*p* < 0.05) of MDA were recorded in *m. gluteobiceps* meat samples. The 10% addition of flaxseed increased MDA values to 0.359 mg.kg^−1^ after 7 days of storage. In contrast, samples of *m. longissimus dorsi* reached only 0.276 mg.kg^−1^. The highest value of MDA (0.325 mg.kg^−1^) was recorded in the control group in *m. longissimus dorsi* after 7 days of storage. Higher values of MDA measured in *m. gluteobiceps* were assumed, because samples from this body area usually contain higher levels of intramuscular fat. In recent years, the scientific trend has been to improve the quality and nutritional value of pork by controlling the intramuscular fat deposition and its fatty acid profile. The nutritional and storage properties are limited due to the low levels of beneficial *n*-3 PUFAs, including long-chain EPA and DHA [55,61]. However, 5% flaxseed supplementation resulted in similar MDA content in meat of the control group. On the other hand, 10% supplementation of flaxseed for 6 weeks did not cause negative effects on human health, because MDA formation was substantial, while the fatty acid profile was significantly improved compared to the control group.

Several nutrition strategies are focused on MDA reduction using antioxidants as a feed supplement to improve the quality of fat. Further research on PUFAs’ protection against oxidation entails the use of vitamin E. Studies of Kim et al. [23] and Huang et al. [32] demonstrated that supplementation of vitamin E (up to 700 IU/kg) can reduce lipid peroxidation in fresh muscle and frozen loin samples. In contrast, Huang et al. [32] noticed that supplementation of vitamin E did not affect MDA values in loin samples. Guo et al. [62] reported that vitamin E can reduce the percentage of saturated fatty acids and increase the proportion of unsaturated fatty acids. Huang et al. [52] confirmed a minimal effect of vitamin E on the fatty acid profile. It is well described that tissue fatty acid profile is influenced more by the type of feed than by the level of feeding [63]. The possibility of antioxidant application in order to reduce levels of MDA when plant oils are used in animal nutrition should be further explored.

## 5. Conclusions

According to our results, we can conclude that 5% and 10% supplementation of flaxseed into commercial feed mixtures for pig fattening significantly affected the composition and quality of produced pork meat. The addition of flaxseed positively influenced the proportion of significant PUFAs, which are beneficial in human nutrition. We observed a change in the FA profile, as well as an increase in the proportions of LA, ALA and EPA in the experimental groups. The proportion of *n*-3 PUFAs in the examined meat samples was significantly increased (*p* < 0.05). The *n*-6/*n*-3 ratio was improved in samples after the administration of 10% flaxseed for 6 weeks compared to the control group. Significantly lower changes in fatty acid profile were observed after 5% supplementation of flaxseed during a 3-week period. In terms of the oxidative stability, we achieved the best results in the F5W3 group. Based on our results, we can recommend the administration of a higher concentration of flaxseed for a shorter period of time to be more appropriate for use in pig diets (F10W3), or a lower concentration of flaxseed for a longer period of feeding (F5W6).

## Figures and Tables

**Table 1 animals-12-01087-t001:** Weight values of pigs during fattening period and after slaughter expressed as carcass yield (means ± SD) expressed as kg^−1^.

	1st Weighting	2nd Weighting	3rd Weighting	Carcass Yield
**C**	73.3 ± 5.16	91.6 ± 4.08	112.1 ± 10.18	87.1 ± 10.18
**F5W3**	78.3 ± 12.54	90.1 ± 3.53	116.8 ± 6.30	91.8 ± 6.30
**F10W3**	76.5 ± 8.95	90.0 ± 3.16	125.5 ± 6.94	100.5 ± 6.94
**F5W6**	75.6 ± 6.37	87.5 ± 8.21	117.1 ± 10.00	92.1 ± 10.01
**F10W6**	79.1 ± 11.58	88.3 ± 9.30	119.1 ± 9.78	94.1 ± 9.78

C: control group fed with commercial complete feed mixture; F5W3: experimental group fed with a diet supplemented with 5% flaxseed for 3 weeks of fattening; F5W6: experimental group fed with a diet supplemented with 5% flaxseed for 6 weeks of fattening; F10W3: experimental group fed with a diet supplemented with 10% flaxseed for 3 weeks of fattening; F10W6: experimental group fed with a diet supplemented with 10% flaxseed for 6 weeks of fattening.

**Table 2 animals-12-01087-t002:** Nutritional composition of standard feed, flaxseed and combination calculated to 100% of dry matter.

Variable	SFM	Flaxseed	SFM + 5% FS	SFM + 10% FS
Crude protein, g/kg	140.42	227.6	145.88	150.51
Crude fat, g/kg	18.07	307.25	37.28	42.23
Crude fiber, g/kg	35.69	232.58	49.60	68.29
NDF, g/kg	176.11	411.9	180.25	189.04
ADF, g/kg	42.27	278.9	54.08	74.47
Ash, g/kg	60.12	35.5	50.49	48.30
Starch, g/kg	574.62	42.32	550.04	510.28
Ca, g/kg	8.12	2.81	7.93	12.80
Mg, g/kg	3.35	4.11	3.36	3.59
Na, g/kg	1.45	4.33	1.34	1.52
K, g/kg	5.58	8.12	5.60	6.29
P, g/kg	4.80	2.71	5.49	5.84
Cu, mg/kg	40.82	33.55	36.50	57.40
Zn, mg/kg	131.83	48.7	104.34	103.45
Mn, mg/kg	150.57	43.83	129.31	125.80
ME, MJ/kg	13.26	12.87	13.31	13.36

SFM—commercial complete feed mixture; FS—flaxseed; NDF—neutral detergent fiber; ADF—acid detergent fiber; ME—metabolizable energy.

**Table 3 animals-12-01087-t003:** Fatty acid composition of control and experimental diets.

Fatty Acids, %	Control	5% Flaxseed	10% Flaxseed
Myristic acid, C14:0	0.101	0.066	0.050
Palmitic acid, C16:0	14.281	5.654	4.496
Stearic acid, C18:0	1.899	3.256	2.547
Linoleic acid, C18:2*n*-6	55.73	16.659	8.547
Gamma-linolenic acid, C18:3*n*-6	0.09	0.015	0.018
Alfa-linolenic acid, C18:3*n*-3	7.109	54.235	72.546
Arachidonic acid, C20:4*n*-6	0.691	0.002	0.052
Eicosapentaenoic acid, C20:5*n*-3	0.125	0.00001	0.0003
Docosapentanoic acid, C22:5*n*-6	0.245	0.011	0.022
Docosapentaenoic acid, C22:5*n*-3	0.043	0.103	0.140
Docosahexaenoic acid, C22:6*n*-3	0.073	0.035	0.055
∑ *n*-3	7.349	54.373	72.741
∑ *n*-6	57.227	16.739	8.720
*n*-6/*n*-3	7.787	0.308	0.120
EPA/AA	0.181	0.0004	0.005
*n*-3 index	7.349	54.373	72.741

EPA—eicosapentanoic acid; AA—arachidonic acid.

**Table 4 animals-12-01087-t004:** The chemical composition of *m. gluteobiceps* and *m. longissimus dorsi*.

** *m. gluteobiceps* **	**% Dry Matter**	**% Fat**	**% Water**	**% Protein**	***p*-Value**
C	28.6 ± 0.26 ^a^	6.34 ± 0.16 ^a^	71.6 ± 0.26 ^c^	20.8 ± 0.02 ^a^	0.044
F5W3	32.1 ± 0.50 ^c^	9.47 ± 0.78 ^b^	68.9 ± 0.50 ^a^	21.9 ± 0.23 ^b^	0.008
F5W6	30.3 ± 1.18 ^b^	8.17 ± 0.82 ^b^	69.7 ± 1.18 ^a^	21.5 ± 0.29 ^b^	0.048
F10W3	33.4 ± 0.12 ^c^	10.84 ± 0.69 ^c^	66.6 ± 0.12 ^b^	20.8 ± 0.03 ^a^	0.045
F10W6	30.8 ± 0.22 ^b^	8.2 ± 0.12 ^b^	69.2 ± 0.22 ^a^	21.7 ± 0.16 ^b^	0.011
** *m. longissimus dorsi* **					
C	29.3 ± 0.18 ^b^	5.5 ± 0.33 ^b^	70.7 ± 0.18 ^c^	22.1 ± 0.42	0.025
F5W3	32.0 ± 0.92 ^c^	7.6 ± 0.27 ^b^	68.0 ± 0.92 ^b^	22.7 ± 0.67	0.048
F5W6	28.4 ± 0.47 ^a^	4.8 ± 0.63 ^a^	71.6 ± 0.47 ^c^	23.0 ± 0.21	>0.05
F10W3	28.4 ± 0.23 ^b^	4.8 ± 0.21 ^a^	71.6 ± 0.23 ^c^	22.3 ± 0.16	0.012
F10W6	34.3 ± 0.46 ^c^	9.7 ± 0.78 ^c^	65.7 ± 0.46 ^a^	22.4 ± 0.81	0.042

C: control group fed with commercial complete feed mixture; F5W3: experimental group fed with a diet supplemented with 5% flaxseed for 3 weeks of fattening; F5W6: experimental group fed with a diet supplemented with 5% flaxseed for 6 weeks of fattening; F10W3: experimental group fed with a diet supplemented with 10% flaxseed for 3 weeks of fattening; F10W6: experimental group fed with a diet supplemented with 10% flaxseed for 6 weeks of fattening. The means with the different superscript letters (a, b, c) in columns are statistically significantly different in individual parameters (Tukey’s post hoc test, *p* < 0.05).

**Table 5 animals-12-01087-t005:** Lipid oxidation determined as MDA (malondialdehyde) expressed as mg.kg^−1^.

** *m. gluteobiceps* **	**Day 1**	**Day 5**	**Day 7**	***p*-Value**
C	0.07 ± 0.01 ^b^	0.10 ± 0.02 ^c^	0.29 ± 0.04 ^b^	0.038
F5W3	0.07 ± 0.01 ^b^	0.12 ± 0.02 ^c^	0.25 ± 0.01 ^b^	0.026
F5W6	0.09 ± 0.01 ^a^	0.20 ± 0.01 ^b^	0.39 ± 0.09 ^a^	0.042
F10W3	0.09 ± 0.01 ^a^	0.27 ± 0.01 ^a^	0.36 ± 0.04 ^a^	0.012
F10W6	0.11 ± 0.02 ^a^	0.28 ± 0.01 ^a^	0.37 ± 0.01 ^a^	0.025
** *m. longissimus dorsi* **				
C	0.08 ± 0.01 ^b^	0.15 ± 0.01 ^b^	0.23 ± 0.13 ^c^	0.001
F5W3	0.09 ± 0.02 ^b^	0.16 ± 0.08 ^b^	0.21 ± 0.09 ^c^	0.015
F5W6	0.13 ± 0.02 ^a^	0.25 ± 0.02 ^a^	0.30 ± 0.02 ^a,b^	0.025
F10W3	0.10 ± 0.01 ^a,b^	0.27 ± 0.01 ^a^	0.28 ± 0.02 ^b^	0.045
F10W6	0.14 ± 0.03 ^a^	0.29 ± 0.03 ^a^	0.33 ± 0.02 ^a^	0.036

C: control group fed with commercial complete feed mixture; F5W3: experimental group fed with a diet supplemented with 5% flaxseed for 3 weeks of fattening; F5W6: experimental group fed with a diet supplemented with 5% flaxseed for 6 weeks of fattening; F10W3: experimental group fed with a diet supplemented with 10% flaxseed for 3 weeks of fattening; F10W6: experimental group fed with a diet supplemented with 10% flaxseed for 6 weeks of fattening. The means with the different superscript letters (a, b, c) in columns are statistically significantly different in individual parameters (Tukey’s post hoc test, *p* < 0.05).

**Table 6 animals-12-01087-t006:** Fatty acid composition of *m. gluteobiceps* fat (means ± SD).

Fatty Acids, %	C	F5W6	F10W6	F5W3	F10W3	*p*-Value
Myristic acid, C14:0	2.97 ± 0.84	2.71 ± 0.77	3.18 ± 0.19	2.85 ± 0.75	2.41 ± 1.08	>0.05
Palmitic acid, C16:0	18.66 ± 4.33	23.37 ± 0.78	24.71 ± 0.67	20.17 ± 4.17	23.63 ± 1.25	>0.05
Stearic acid, C18:0	4.88 ± 0.81	4.83 ± 0.18	4.80 ± 0.11	5.49 ± 0.62	4.96 ± 0.21	>0.05
Linoleic acid, C18:2*n*-6	31.33 ± 1.77 ^a^	20.28 ± 3.73 ^b^	12.39 ± 0.79 ^c^	21.52 ± 4.26 ^b^	12.98 ± 0.42 ^c^	0.035
Gamma-linolenic acid, C18:3*n*-6	0.05 ± 0.006	0.05 ± 0.007	0.04 ± 0.006	0.06 ± 0.009	0.05 ± 0.007	>0.05
Alfa-linolenic acid, C18:3*n*-3	1.73 ± 0.51 ^c^	2.64 ± 0.64 ^b^	7.16 ± 1.12 ^a^	2.23 ± 0.34 ^b^	2.79 ± 1.02 ^b^	0.022
Arachidonic acid, C20:4*n*-6	2.50 ± 0.316	2.58 ± 0.603	2.36 ± 0.049	2.54 ± 0.337	2.59 ± 0.319	>0.05
Eicosapentaenoic acid, C20:5*n*-3	0.05 ± 0.005 ^c^	0.10 ± 0.026 ^b^	0.16 ± 0.008 ^a^	0.05 ± 0.005 ^c^	0.06 ± 0.018 ^c^	<0.001
Docosapentanoic acid, C22:5*n*-6	0.32 ± 0.042 ^a^	0.32 ± 0.105 ^a^	0.19 ± 0.008 ^b^	0.39 ± 0.044 ^a^	0.19 ± 0.079 ^b^	<0.001
Docosapentaenoic acid, C22:5*n*-3	0.43 ± 0.046 ^c^	0.68 ± 0.042 ^b^	0.76 ± 0.016 ^a^	0.45 ± 0.052 ^c^	0.40 ± 0.043 ^c^	0.002
Docosahexaenoic acid, C22:6*n*-3	1.72 ± 0.27 ^c^	3.36 ± 0.38 ^a^	2.84 ± 0.07 ^ab^	2.48 ± 0.35 ^b^	1.70 ± 0.26 ^c^	0.005
∑*n*-3	3.93 ± 0.68 ^c^	6.78 ± 0.44 ^b^	10.92 ± 1.07 ^a^	5.20 ± 0.36 ^b^	4.958 ± 0.86 ^b^	0.046
∑*n*-6	34.15 ± 1.95 ^a^	23.23 ± 4.83 ^b^	14.88 ± 0.81 ^c^	24.51 ± 4.61 ^b^	15.807 ± 0.74 ^c^	0.031
*n*-6/*n*-3	8.69 ± 0.98 ^a^	3.42 ± 0.93 ^b^	1.36 ± 0.20 ^c^	4.71 ± 1.22 ^b^	3.18 ± 0.77 ^b^	0.024
EPA/AA	0.02 ± 0.002 ^c^	0.04 ± 0.027 ^b^	0.07 ± 0.007 ^a^	0.02 ± 0.002 ^c^	0.02 ± 0.021 ^b,c^	0.015
*n*-3 index	3.93 ± 0.68 ^c^	6.78 ± 0.44 ^b^	10.92 ± 1.07 ^a^	5.20 ± 0.368 ^b^	4.95 ± 0.85 ^b^	0.020

C: control group fed with commercial complete feed mixture; F5W3: experimental group fed with a diet supplemented with 5% flaxseed for 3 weeks of fattening; F5W6: experimental group fed with a diet supplemented with 5% flaxseed for 6 weeks of fattening; F10W3: experimental group fed with a diet supplemented with 10% flaxseed for 3 weeks of fattening; F10W6: experimental group fed with a diet supplemented with 10% flaxseed for 6 weeks of fattening. EPA—eicosapentanoic acid; AA—arachidonic acid. The means with the different superscript letters (a, b, c) in rows are statistically significantly different (Tukey’s post hoc test, *p* < 0.05).

**Table 7 animals-12-01087-t007:** Fatty acid composition of *m. longissimus dorsi* fat (means ± SD).

Fatty Acids, %	C	F5W6	F10W6	F5W3	F10W3	*p*-Value
Myristic acid, C14:0	2.61 ± 0.65	1.60 ± 0.82	2.36 ± 0.71	2.53 ± 0.71	2.37 ± 0.69	>0.05
Palmitic acid, C16:0	24.10 ± 0.80 ^a^	22.04 ± 0.98 ^b^	22.64 ± 0.81 ^b^	24.05 ± 0.86 ^a^	24.89 ± 1.84 ^a^	>0.05
Stearic acid, C18:0	14.67 ± 0.20 ^b^	16.88 ± 0.82 ^a^	16.76 ± 1.51 ^a^	16.98 ± 1.223 ^a^	15.83 ± 1.257 ^a,b^	>0.05
Linoleic acid, C18:2*n*-6	27.76 ± 1.51 ^a^	23.35 ± 1.11 ^b^	18.20 ± 0.89 ^c^	19.24 ± 2.65 ^c^	19.22 ± 1.433 ^c^	0.017
Gamma-linolenic acid, C18:3*n*-6	0.06 ± 0.009	0.18 ± 0.011 ^a,^***	0.06 ± 0.006 ^b^	0.08 ± 0.034 ^a^	0.12 ± 0.018 ^a,^**	>0.05
Alfa-linolenic acid, C18:3 *n*-3	1.03 ± 0.26 ^c^	2.89 ± 0.47 ^b^	5.50 ± 0.54 ^a^	2.00 ± 0.15 ^b^	2.66 ± 0.19 ^b^	0.007
Arachidonic acid, C20:4 *n*-6	0.93 ± 0.13 ^d^	8.31 ± 0.81 ^a^	5.29 ± 0.23 ^b^	2.52 ± 1.04 ^c^	9.36 ± 0.38 ^a^	>0.05
Eicosapentaenoic acid, C20:5*n*-3	0.05 ± 0.007 ^d^	0.96 ± 0.175 ^a^	0.84 ± 0.049 ^a^	0.10 ± 0.020 ^c^	0.34 ± 0.095 ^b^	0.010
Docosapentanoic acid, C22:5*n*-6	0.10 ± 0.013 ^c^	0.15 ± 0.090 ^b^	0.09 ± 0.058 ^c^	0.16 ± 0.023 ^b^	0.30 ± 0.034 ^a^	0.009
Docosapentaenoic acid, C22:5*n*-3	0.11 ± 0.028 ^e^	1.16 ± 0.115 ^b^	1.51 ± 0.034 ^a^	0.35 ± 0.090 ^d^	0.66 ± 0.097 ^c^	0.003
Docosahexaenoic acid, C22:6*n*-3	0.36 ± 0.044 ^d^	3.80 ± 0.241 ^a^	2.24 ± 0.071 ^b^	0.82 ± 0.265 ^c^	2.50 ± 0.104 ^b^	<0.001
∑*n*-3	1.54 ± 0.234 ^e^	8.83 ± 0.52 ^b^	10.11 ± 0.55 ^a^	3.27 ± 0.29 ^d^	6.17 ± 0.29 ^c^	0.003
∑*n*-6	28.84 ± 1.41 ^a^	31.93 ± 2.22 ^a^	23.63 ± 1.18 ^b^	22.01 ± 3.69 ^b^	29.01 ± 1.53 ^a^	0.006
*n*-6/*n*-3	18.75 ± 2.38 ^a^	3.61 ± 0.48 ^c^	2.34 ± 0.25 ^d^	6.73 ± 0.61 ^b^	4.69 ± 0.42 ^c^	0.005
EPA/AA	0.05 ± 0.015 ^c^	0.12 ± 0.023 ^b^	0.16 ± 0.004 ^a^	0.04 ± 0.010 ^c^	0.04 ± 0.013 ^c^	0.032
*n*-3 index	1.53 ± 0.23 ^e^	8.83 ± 0.52 ^b^	10.11 ± 0.54 ^a^	3.27 ± 0.29 ^d^	6.17 ± 0.29 ^c^	0.001

C: control group fed with commercial complete feed mixture; F5W3: experimental group fed with a diet supplemented with 5% flaxseed for 3 weeks of fattening; F5W6: experimental group fed with a diet supplemented with 5% flaxseed for 6 weeks of fattening; F10W3: experimental group fed with a diet supplemented with 10% flaxseed for 3 weeks of fattening; F10W6: experimental group fed with a diet supplemented with 10% flaxseed for 6 weeks of fattening. EPA—eicosapentanoic acid; AA—arachidonic acid. The means with the different superscript letters (a, b, c, d, e) in rows are statistically significantly different (Tukey’s post hoc test, *p* < 0.05); **—statistically significant compared to control group (*p* < 0.01); ***—statistically significant compared to control group (*p* < 0.001).

## Data Availability

Not applicable.

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
