# Peer review of "Effect of Concentration of Flaxseed (*Linum usitatissimum*) and Duration of Administration on Fatty Acid Profile, and Oxidative Stability of Pork Meat"

_animals, 2022, doi:10.3390/ani12091087_

Round 1

Reviewer 1 Report

The topic of the paper deals with an interesting and relevant question, and I believe that the study is relevant and fits well into the general framework of the Animals journal. However, the paper needs some necessary improvements.

Let me briefly highlight the issues that need to be corrected:

Materials and Methods:

L: 114 – 115: the Authors pointed out the composition of the experimental diets, which is given in Table 1. However, the Authors did not provide any information on the methods of establishing the nutritional and energy value of the complete mixtures used. The question is: did the Authors take this data from the supplier of the complete feed mixture, or did they perform the appropriate AOAC analyzes, and then made a recalculations?

In the L: 127 – 133, the Authors gave some information’s on performing AOAC procedures on the meat of pigs, but typically the same procedures should be undertaken to evaluate the proper amounts of nutritional ingredients in the used mixtures. Similarly, these analyzes should be performed for flaxseed, in order estimate the nutritional and energetic value of used experimental feed mixtures.

Also, there is a lack of information’s about the nutritional requirements of pigs used for calculation of the metabolic energy (ME), based on the chemical composition of the feed.

The rest of the methodology section is described correctly and I have no further comments in this matter.

Discussion:

The discussion section is very extensive and its shortening should be considered.

1st: the paragraph given in the L: 400 – 415 is not necessary, as it is not directly linked with the topic of the paper and the aim of this study. Instead of this part, one sentence regarding the possibility of antioxidant application in order to reduce levels of MDA, when plant oils are used in animal nutrition (which is a common practice).

2nd: the part relating to oxidation given in the L: 273 – 278 is out of context and is completely not related to discussed issues (proportion of FA, energy metabolism, etc.). This fragment seems to be continued from the L: 363, where the influence of extended PUFA’s (especially n-3) on MDA is discussed.

3rd: in the L: 389 – 390 the Authors indicated: “Surprisingly the highest value of MDA (0.432 mg.kg-1) was recorded in control group in  m. longissimus dorsi after 7 days of storage.”

Typically, this results should be related to some procedural artefacts or the specifics of the tested sample (then a repeat of the procedure could confirm or reject the controversial results). On the other hand, the higher value of MDA in control samples after 7 d of storage may be related to a “false negative” result, which may be a consequence of reduction the amount of PUFAs (due to accelerated oxidation related to the presence of n-3 FA) in the other samples of meat. When peroxidation terminates or is near to termination, the measurable amounts of MDA are lower than in samples which are les oxidized (for example the control, due to higher stability), which can give higher values of MDA. However, the problem with this sentence is the value of 0.432 mg / kg, which does not correspond with the value 0.325 mg / kg given in Table 3. Also, the value 0.311 mg / kg is not included in the Table, so I suggest to review the L: 388 – 389 to check these values.

In other parts I have no additional comments. The manuscript is written reasonably well, so the only the necessary corrections given above needs to be applied.

Best wishes,

Author Response

  1. L: 114 – 115: the Authors pointed out the composition of the experimental diets, which is given in Table 1. However, the Authors did not provide any information on the methods of establishing the nutritional and energy value of the complete mixtures used. The question is: did the Authors take this data from the supplier of the complete feed mixture, or did they perform the appropriate AOAC analyzes, and then made a recalculations?

Commercial compound feed data were collected according to the supplier of the selected commercial feeds. The values of compound were measured and recalculated at the Department of Dietetics.

  1. In the L: 127 – 133, the Authors gave some information’s on performing AOAC procedures on the meat of pigs, but typically the same procedures should be undertaken to evaluate the proper amounts of nutritional ingredients in the used mixtures. Similarly, these analyzes should be performed for flaxseed, in order estimate the nutritional and energetic value of used experimental feed mixtures.

Flaxseed was measured and added to commercial feed mixtures as required by the experiment. Subsequently, the feed mixtures were examined and supplemented with individual components in order to achieve the same energy and nutritional balance of experimental feeds.

  1. Also, there is a lack of information’s about the nutritional requirements of pigs used for calculation of the metabolic energy (ME), based on the chemical composition of the feed.

Nutritional requirements have been recalculated based on our national standards for pig farming.

  1. The rest of the methodology section is described correctly and I have no further comments in this matter.

Discussion:

The discussion section is very extensive and its shortening should be considered.

  1. 1st: the paragraph given in the L: 400 – 415 is not necessary, as it is not directly linked with the topic of the paper and the aim of this study. Instead of this part, one sentence regarding the possibility of antioxidant application in order to reduce levels of MDA, when plant oils are used in animal nutrition (which is a common practice).

The section has been re-evaluated and rewritten as required.

  1. 2nd: the part relating to oxidation given in the L: 273 – 278 is out of context and is completely not related to discussed issues (proportion of FA, energy metabolism, etc.). This fragment seems to be continued from the L: 363, where the influence of extended PUFA’s (especially n-3) on MDA is discussed.

This section has been moved and rewritten on the oxidation of PUFAs.

  1. 3rd: in the L: 389 – 390 the Authors indicated: “Surprisingly the highest value of MDA (0.432 mg.kg-1) was recorded in control group in   longissimus dorsi after 7 days of storage.” Typically, this results should be related to some procedural artefacts or the specifics of the tested sample (then a repeat of the procedure could confirm or reject the controversial results). On the other hand, the higher value of MDA in control samples after 7 d of storage may be related to a “false negative” result, which may be a consequence of reduction the amount of PUFAs (due to accelerated oxidation related to the presence of n-3 FA) in the other samples of meat. When peroxidation terminates or is near to termination, the measurable amounts of MDA are lower than in samples which are les oxidized (for example the control, due to higher stability), which can give higher values of MDA. However, the problem with this sentence is the value of 0.432 mg / kg, which does not correspond with the value 0.325 mg / kg given in Table 3. Also, the value 0.311 mg / kg is not included in the Table, so I suggest to review the L: 388 – 389 to check these values.

Values were reviewed and rewritten according to reviewer’s statement.

Reviewer 2 Report

The article presented for review concerns the effect of feed supplementation given to pigs on the fatty acid profile and oxidative stability of meat. The research seems to be quite interesting, but not very innovative.

In the opinion of the Reviewer, the manuscript is not suitable for printing because it has many serious disqualifying errors:

  1. The feed given to pigs was not properly balanced.

Table 1 shows that the feed in all variants with flax additives has the same calorific value. How is it possible that the addition of 10% flax increased the amount of fat in the feed 2 times (and the feed base was the same in each variant!), and not increased the amount of energy? After all, 1g of fat is 9kcal. This raises serious doubts about the pig breeding, the course of the experiment and the interpretation of the obtained results! This setting of experience is wrong.

  1. Did such a large addition of flax seeds to the feed cause a laxative effect?

In the Reviewer's opinion, such an additive may have an impact on animal health, which means that the research should have the consent of the local ethics committee to conduct the research described in the manuscript.

  1. The authors obtained an increase in the amount of unsaturated fatty acids, but what about the taste and smell of such meat? Was it acceptable?

The fact that the increase in desired parameters (PUFA) has been obtained does not necessarily mean that the raw material or the manufactured product will be sensorically acceptable. The manuscript should be supplemented with data from the organoleptic analysis.

 4. In the case of determining the oxidative stability of fat, the authors explain the obtained relationships only with the type of enriched feed administered and the storage time. What about bacterial spoilage changes?

The study should be supplemented with the results of changes in microbial contamination measured with the total amount of bacteria, the amount of Pseudomonas bacteria, the amount of lactic acid bacteria and the amount of Enterobacteriaceae bacteria.

  1. In the opinion of the Reviewer, the biggest drawback of the study is the insufficient number of individuals in the experimental groups and the lack of information on the sex of the animals.

In the opinion of the reviewer, such a small number of individuals in the groups (or the lack of repetitions) makes it impossible to obtain reliable results!

  1. References are prepared exceptionally carelessly.

Editorial errors in references 1, 2, 3, 8, 10, 12, 13, 16, 20, 22, 24, 25, 28, 29, 31, 34, 36, 39, 56, 60, 64.

Author Response

Dear reviewer, first of all thank you for your time, willingness and for your valuable comments, which helped us to improve the overall quality of the revised manuscript. All changes and modifications were highlighted with yellow color to indicate changes we have made in the revised text. We used professional English editing institution to improve the manuscript significantly.

  1. The feed given to pigs was not properly balanced.

Table 1 shows that the feed in all variants with flax additives has the same calorific value. How is it possible that the addition of 10% flax increased the amount of fat in the feed 2 times (and the feed base was the same in each variant!), and not increased the amount of energy? After all, 1g of fat is 9kcal. This raises serious doubts about the pig breeding, the course of the experiment and the interpretation of the obtained results! This setting of experience is wrong.

Values of feed energy and caloric values were measured before feeding. Feed mixtures were recalculated  and re-aranged for the equality of experimental feed mixtures compared to control diet.

  1. Did such a large addition of flax seeds to the feed cause a laxative effect?

In the Reviewer's opinion, such an additive may have an impact on animal health, which means that the research should have the consent of the local ethics committee to conduct the research described in the manuscript.

The study was evaluated by the ethics committee before the start of the experiment and was approved. Approval by the ethics committee was added to the submission of the article. The animals used in the experiment were under the supervision of the pig clinic and the department of physiology. No laxative effects were observed.

  1. The authors obtained an increase in the amount of unsaturated fatty acids, but what about the taste and smell of such meat? Was it acceptable?

The fact that the increase in desired parameters (PUFA) has been obtained does not necessarily mean that the raw material or the manufactured product will be sensorically acceptable. The manuscript should be supplemented with data from the organoleptic analysis.

Produced meat was sensory evaluated by a panel of evaluators at the Department of Technology Hygiene and Food Safety of the University of Veterinary medicine and pharmacy in Košice. There was no change in terms of sensory and organoleptic properties (data not presented)

  1. In the case of determining the oxidative stability of fat, the authors explain the obtained relationships only with the type of enriched feed administered and the storage time. What about bacterial spoilage changes?

The study should be supplemented with the results of changes in microbial contamination measured with the total amount of bacteria, the amount of Pseudomonas bacteria, the amount of lactic acid bacteria and the amount of Enterobacteriaceae bacteria.

This research did not focus on microbial spoilage of meat. All samples were taken directly at the slaughterhouse and vacuum packed for refrigerator storage.

  1. In the opinion of the Reviewer, the biggest drawback of the study is the insufficient number of individuals in the experimental groups and the lack of information on the sex of the animals.

The selected number of individual animals was determined as the minimum necessary for a statistically significant evaluation of the experimental results. The sex of the animals was added to the methodology

  1. References are prepared exceptionally carelessly.

Editorial errors in references 1, 2, 3, 8, 10, 12, 13, 16, 20, 22, 24, 25, 28, 29, 31, 34, 36, 39, 56, 60, 64.

The references were rewritten and modified according to the requirements of the reviewer and editor

We believe that this revised manuscript is appropriate for publication.

Yours sincerely,

team of authors.

Reviewer 3 Report

Effect of Concentration of Flaxseed (Linum usitatissimum) and Length of Administration on Fatty Acid Profile, and Oxidative stability of Pork Meat

The authors have studied the effect of flaxseed on the fat content and fatty acid composition of MLD and MGB muscles over 6 or 3 weeks before slaughter (not clearly described in material and methods). It is also not clear; how much muscles and adipose tissues were considered. Sex of pigs not mentioned. The performance data before and during the experiments are completely missing. (When was the 3 weeks treatment started relative to the 6 weeks treatment?). The results seem not homogenous. E.g., Tab. 2: fat content in F10W3 and F19W6 or in Tab 5: DHA in F10W6 and F10W3. Etc. It is difficult to trust the results in general.

Some statements for improvement of the paper:

  • Simple Summary: Pig meat is not a main source of SFA. The improvement of the n-3/n-6 relation is another aspect. Health aspects and sensory quality of pork (oxidation) must be considered very carefully.
  • Abstract: Flaxseed is not one of the most common feed supplements in livestock! But it has special properties that can be highly relevant in pig feeding.
  • 39: versus what?
  • 1: Are these analyzed or calculated values? Add values for Flaxseed used in the experiment
  • 153 ff: Table on pig performance before and during experiment are missing. They must be added in the manuscript!
  • 2: Delete DM content: 100 % minus DM % is equal water %
  • Reduce in all Tabs. the number of digits to a informative number.

Author Response

Dear reviewer, first of all thank you for your time, willingness and for your valuable comments, which helped us to improve the overall quality of the revised manuscript. All changes and modifications were highlighted with yellow color to indicate changes we have made in the revised text. We used professional English editing institution to improve the manuscript significantly.

  1. The authors have studied the effect of flaxseed on the fat content and fatty acid composition of MLD and MGB muscles over 6 or 3 weeks before slaughter (not clearly described in material and methods). It is also not clear; how much muscles and adipose tissues were considered. Sex of pigs not mentioned. The performance data before and during the experiments are completely missing. (When was the 3 weeks treatment started relative to the 6 weeks treatment?). The results seem not homogenous. E.g., Tab. 2: fat content in F10W3 and F19W6 or in Tab 5: DHA in F10W6 and F10W3. Etc. It is difficult to trust the results in general.

The complaines about the study was added to the methodology. The sex of the animals has been added as requested. The average weights of the animals were weighted at the beginning of the experiment, during experiment and at the end of slaughter. The weight values were added as a new table to the manuscript

Some statements for improvement of the paper:

  1. Simple Summary: Pig meat is not a main source of SFA. The improvement of the n-3/n-6 relation is another aspect. Health aspects and sensory quality of pork (oxidation) must be considered very carefully.

Meat and meat products are considered to be the main source of dietary fats, especially as a source of saturated fatty acids  we did not mention pig meat as a main source of SFA, but meat and meat products in general.

  1. Abstract: Flaxseed is not one of the most common feed supplements in livestock! But it has special properties that can be highly relevant in pig feeding.

Abstract was redesigned according to the reviewers.

  1. 39: versus what?

versus control group, Thank you I will add this information into the manuscript.

  1. 1: Are these analyzed or calculated values? Add values for Flaxseed used in the experiment

Values of flaxseed were added. The values in table 1. Were analysed.

  1. 153 ff: Table on pig performance before and during experiment are missing. They must be added in the manuscript!

Table of pig performance was added.

  1. 2: Delete DM content: 100 % minus DM % is equal water %

We consider that the values of DM are important.

  1. Reduce in all Tabs. the number of digits to a informative number.

Number were reduced to an informative number according to the another reviewer.

We believe that this revised manuscript is appropriate for publication.

Yours sincerely,

team of authors.

Reviewer 4 Report

Reviewing the manuscript animals-1650158

Effect of Concentration of Flaxseed (Linum usitatissimum) and 2 Length of Administration on Fatty Acid Profile, and Oxidative stability of Pork Meat

Relevance and Information content:

This study was conducted to investigate the impact of flaxseed on pork meat quality The topic of the manuscript is of interest of the audience of the journal. The manuscript is of very good English language quality. However, there seems to be confusion about the understanding of the fatty acid profile, quantities and ratio in diets, detailed comments are given below. In addition, Mat & meth are lacking relevant information which needs to be added. Analysis would have benefitted substantially from 3x2 factorial design taking 3 flaxseed inclusion levels (0, 5, 10%) and 2 feeding durations (3, 6 weeks) into account to investigate main effects and interaction.

Title: please replace “length of administration” with “duration of administration”.

Simple summary:

Line 19: please reserve “significantly” to statistically significant differences only. Therefore in this case, please replace with “substantially” or similar.

Line 19: Higher in Western countries compared to what?

Line 19-20: I assume you are referring to human fat intake, not pig’s fat intake? Given that you are aiming to publish in “animals”, please be clear.

Line 20: I assume you refer to change of human food consumption or pig food consumption? Please clarify

Line 24: please change to: Flaxseed with 24 57% alpha-linolenic acid (ALA) concentration was used in the diet of swine.

Line 25: please replace “concentrations” to “inclusion levels…. fed at two time intervals… ”.

Abstract:

Line 31: Flaxseed is mostly used as feed ingredient rather than supplement. Please change

Line 31: most common is only true for some countries. Which countries do you refer to? Please state

Line 31: Were the flaxseed nutrients taken into account for diet formulation or was flaxseed added over the top? Only in case of the latter the word “supplement” would be correct to be used here. Based on Table 1 that information is not clear.

Line 33: what were the intervals? Or was the flaxseed given for the duration of 3 and 6 weeks prior slaughter?

Line 34: What are the 5 groups? Please state the experimental arrangement and statistical design.

Line 43: related to

Introduction:

Line 49: please quantify “most consumers” and specify which countries you are referring to.

Line 50-51: should be moved to the end of the introduction, first focus on the relevance on the topic, outline the problem, THEN how this study aims to contribute solving the problem.

Line 52: replace “of n-3 and n-6 line” with “with n-3 and n-6 characteristics”

Line 54: the human and pig organism uses NOT the same enzyme complex for the metabolism of these fatty acids, which is why the have different biological implications e.g. modulate inflammatory pathways differently, resulting in different health conditions. This is why the absolute intake of n 3, n 6 and the ration of n3:n6 is important. Please correct.

Line 59: what recommendations exactly? Made by whom? Please be specific and precise.

Line 60: which population? Pig population? In what country? In what year?

Line 61: how much below what limit in what countries? You are writing for a global audience here> Be specific and precise.

Line 63: “higher content” of LA and ALA compared to what?

Line 63: Animal fats and vegetable oils have a very different fatty acid profile and very distinct levels of LA and ALA, why are they mentioned here together? Fish oils (which is an animal fat) provide highest rates of EPA and DHA, so this sentence makes no sense.

Line 63: in line 55 you outlined the importance of the essential fatty acids LA and ALA, here you are saying the “higher” content of LA and ALA is not good. You are contradicting yourself, please re-phrase to clarify.  

Line 67: there is a difference between the required/ideal quantity of n3 and n 6 fats, and the ideal n3/n6 ration. Do not mix up the two. You should introduce the reader to both parameters, don’t mix them up as you did here. Your ratio can be good while you not meet sufficient quantity and vice versa. What is the ratio and absolute values I pork meat usually and what are the recommendations for humans? Has the idea ration and/or absolute quantity achieved so far in pork using (any) feed ingredients / by any researchers? If not in pork, then in other meat sources? Why are you aiming to modify fatty acid composition in pork and not any other meat? Especially given that pork consumption is experiencing a long ongoing decline in many societies (probably exactly for its “unhealthy” reason)?

Line 68: and what was the outcome of these research experiments?

Line 68: I believe it was not the addition but rather the use of various oils. If you add oil you increase total energy of the diet, which brings many other problems. It is rather than feed formulation is altered to include “new” fat sources e.g. one fat source with a (more beneficial) different fatty acid composition replacing another. So please rephrase and do not use “addition” unless some feed ingredient was added “over the top” of a formulated diet.

Line 68: research has also used flaxseed oil in other species, evening primrose, sunflower oil, thistle oil, fish oils, fish meal. Please include, ideally in table form stating inclusion levels and outcomes.

Line 72-76: what was the inclusion level, duration of feeding and was the result statistically significant? These are only some relevant reasons why results vary so much. Without this context it does not become clear where the gap of knowledge is that your work is trying to fill and t also does not allow the reader to judge your results and critically evaluate your discussion.

Line 79: Please define “natural” antioxidant. Everything in this world comes from nature.

Line 79: antioxidants are given in pig diets ever since pigs evolved. Vitamin E is an antioxidant and required for every pig, uptaken by wild pigs as much as commercial housed pigs. So its not a new approach at all, please be more precise as what you are trying to say.

Line 81: simultaneously administration of PUFA and what? PUFA alone will increase oxidation rate in meat.

Line 83: What specific substances are you referring to? What is your definition of “specific” substance?

Line 83: Please replace “supplemented” for reasons outlined above.

Line 84: The information in this sentence is not correct. Canola oil, soybean oil and linseed oil not only shorty chain fatty acids, but also essential fatty acids which are long chain, they also provide medium chain FA. I highly encourage the authors to critially revise the literatire on dietary fatty acids!

Line 86-90: Based on the information given until here it is not clear:

  • Why the specific concentration of flax seeds are used
  • Why flaxseed was chosen to be investigated (is it a superior feed source over other dietary options, is it cheaper? More available? Etc)
  • What changes in FA profile and ratio to expect (based on previous research, the fatty acid profile of flaxseed and the inclusion level in the diet, the expected concentration in the pig meat can be predicted), therefore making this study a proof of concept.
  • In what ways “intervals” (duration?) would be relevant or had been chosen to be investigated?
  • How antioxidants (line 79) play a role here- based on “simultaneous administration” explanation, why are antioxidants not investigated?

Materials and Methods:

Line 94: please state the animal ethics approval number and authority.

How were the animals assigned to each group? Why was the flaxseed not taken into diet formulation but rather supplemented over the top? This would have impacted energy density and subsequently feed intake of the diet

Table 1: please include detailed information about the diet composition listing all feed ingredients

Please take into consideration the nutrient dilution factor when adding the flaxseed over the top and provide calculated values of the final experimental diet that the pigs consumed  

Table 1 should also show the nutrient composition AS ANALYSED, since this always differs from the calculated composition.

Line 114: Please share the total nutrient and fatty acid profile of the flaxseed.

Table 1 also needs information about the microminerals, of special interest is the vitamin composition as some of them serve as antioxidants)

Section 2.1: missing information about animal housing and killing: light, bedding, temperature, housing system, vaccination status, frequency of health checks, etc. The experiment is not reproducible with the current information given.

Section 2.2 not clear how meat was processed before sample taking: animal transport and handling affects stress and therefore meat quality, stunning and killing method can affect meat quality as well as post- killing processing and time until sample removal. Please outline on all of it.

Statistical Analysis:

What was your experimental unit, how many replicates, what was the statistical arrangement and design, how was data tested for normality, how treated if not normally distributed, how were outliers determined and handled.

Analysis would have benefitted substantially from 3x2 factorial design taking 3 flaxseed inclusion levels (0, 5, 10%) and 2 feeding durations (3, 6 weeks) into account!!! Please change so you are able to investigate main effects and interaction.

Results:

Line 154: “In table 1 is shown…”

No comments

Line 158: replace “mixture” with “diet”

Line 159: negatively is a subjective opinion by the authors, please delete/replace with objective parameters (e.g. increased by xxx). Feeding too much or too little energy will have consequences of the pig owner’s finances to start with, so its all a matter or perspective.

All tables: Apply 3 – digit rule for better readability e.g. xx.x  x.xx

Discussion:

It is really hard to follow the discussion of the results without having the full information on the experimental diets available. Discussion needs to be more differentiated and critical as outlined in my comments on the introduction already. When referring to previous research results it should be clear in which species, at what inclusion level, fed for what duration. Especially since this manuscript covers dietary intake of pigs and humans there are very frequent unclarities, but also information is available on many other animal species which raises questions.  

Author Response

Dear reviewer, first of all thank you for your time, willingness and for your valuable comments, which helped us to improve the overall quality of the revised manuscript. All changes and modifications were highlighted with yellow color to indicate changes we have made in the revised text. We used professional English editing institution to improve the manuscript significantly.

Relevance and Information content:

This study was conducted to investigate the impact of flaxseed on pork meat quality The topic of the manuscript is of interest of the audience of the journal. The manuscript is of very good English language quality. However, there seems to be confusion about the understanding of the fatty acid profile, quantities and ratio in diets, detailed comments are given below. In addition, Mat & meth are lacking relevant information which needs to be added. Analysis would have benefitted substantially from 3x2 factorial design taking 3 flaxseed inclusion levels (0, 5, 10%) and 2 feeding durations (3, 6 weeks) into account to investigate main effects and interaction.

Title: please replace “length of administration” with “duration of administration”.

Title was replaced with “duration of administration”.

Simple summary:

Line 19: please reserve “significantly” to statistically significant differences only. Therefore in this case, please replace with “substantially” or similar.

Words was replaced in an appropriate way.

Line 19: Higher in Western countries compared to what?

Compared to the mediteriann or eastern diet, we know that eastern diet use pork meat, but the main meat for consumption is dominantly poultry and fishes.

Line 19-20: I assume you are referring to human fat intake, not pig’s fat intake? Given that you are aiming to publish in “animals”, please be clear.

We rewrite it more specific to the human consumption of pork meat to be clear.

Line 20: I assume you refer to change of human food consumption or pig food consumption? Please clarify

We rewrite it more specific to the human consumption of pork meat to be clear.

Line 24: please change to: Flaxseed with 24 57% alpha-linolenic acid (ALA) concentration was used in the diet of swine.

Line 24 was rewritten according to another reviewer and it is in present form: “Flaxseed with a concentration of alpha-linolenic acid (ALA) at 57% was used in the swine diet.“

Line 25: please replace “concentrations” to “inclusion levels…. fed at two time intervals… ”.

Line 25 was rewritten: Flaxseed supplementation at two inclusion levels (5 and 10%) in two time intervals (3 and 6 weeks before slaughter) was evaluated

Abstract:

Line 31: Flaxseed is mostly used as feed ingredient rather than supplement. Please change

Line 25 was rewritten: Flaxseed is a common feed ingredient used for livestock

Line 31: most common is only true for some countries. Which countries do you refer to? Please state

We refer to the European countries (Hungary, Poland, Czech Republic, Italy, France, etc.) but also flaxseed oil is used in eastern countries like Thailand and China.

Line 31: Were the flaxseed nutrients taken into account for diet formulation or was flaxseed added over the top? Only in case of the latter the word “supplement” would be correct to be used here. Based on Table 1 that information is not clear.

Flaxseed was added on the top and then mixed properly with commercial diet pellets.

Line 33: what were the intervals? Or was the flaxseed given for the duration of 3 and 6 weeks prior slaughter?

Yes we add the flaxseed into the diet for 3 and 6 weeks before slaughter.

Line 34: What are the 5 groups? Please state the experimental arrangement and statistical design.

We were more specific in the Mat. And Meth. Section for the group division.

Line 43: related to

Thank you, the word was rewritten.

Introduction:

Line 49: please quantify “most consumers” and specify which countries you are referring to.

“The human diet structure of the most consumers from the Europe, does not meet the recommendations of,…..”

Line 50-51: should be moved to the end of the introduction, first focus on the relevance on the topic, outline the problem, THEN how this study aims to contribute solving the problem.

This experiment was focused on improving the quality of pork meat, by supplementation of the pigs’ diet with flaxseed (Linum usitatissimum).This study investigates the supplementation of two different concentrations of flaxseed and its supplementation in various intervals during two fattening periods on the quality of produced pork meat. The changes in the FA profile and the ratio of n-3/n-6 PUFAs on pork meat were investigated

Line 52: replace “of n-3 and n-6 line” with “with n-3 and n-6 characteristics”

“n-3 and n-6 characteristics have the same metabolic rate and… “

Line 54: the human and pig organism uses NOT the same enzyme complex for the metabolism of these fatty acids, which is why the have different biological implications e.g. modulate inflammatory pathways differently, resulting in different health conditions. This is why the absolute intake of n 3, n 6 and the ration of n3:n6 is important. Please correct.

“… (PUFAs) of n-3 and n-6 characteristics have not the same metabolic rate and the organism does not use the same….”

Line 59: what recommendations exactly? Made by whom? Please be specific and precise.

“ Despite all the recommendations of the WHO, the population does not receive sufficient amounts of essential PUFAs [4]. Daily intake of eicosapentaenoic …. “

Line 60: which population? Pig population? In what country? In what year?

“ Despite all the recommendations of the WHO, the human population does not receive sufficient amounts of essential PUFAs [4]. Daily intake of eicosapentaenoic …. “

Line 61: how much below what limit in what countries? You are writing for a global audience here> Be specific and precise.

“….. 5 n-3) and docosahexaenoic acid (DHA, 22:6 n-3) is well below the limit in Western Europe and American country…. “

Line 63: “higher content” of LA and ALA compared to what?

“…. ALA in produced food, where the nutritional trends….”

Line 63: Animal fats and vegetable oils have a very different fatty acid profile and very distinct levels of LA and ALA, why are they mentioned here together? Fish oils (which is an animal fat) provide highest rates of EPA and DHA, so this sentence makes no sense.

The main reason why we mention this is that fish oil is used as an additive and has a high content of EPA and DHA, but its price is relatively high compared to flaxseed or flaxseed oil

Line 63: in line 55 you outlined the importance of the essential fatty acids LA and ALA, here you are saying the “higher” content of LA and ALA is not good. You are contradicting yourself, please re-phrase to clarify.  

We say that a high proportion of LA and ALA is significant, but despite the fact that many countries with a high proportion of LA and ALA which are consumed in Western European countries, consumers still do not receive enough EPA and DHA.

Line 67: there is a difference between the required/ideal quantity of n3 and n 6 fats, and the ideal n3/n6 ration. Do not mix up the two. You should introduce the reader to both parameters, don’t mix them up as you did here. Your ratio can be good while you not meet sufficient quantity and vice versa. What is the ratio and absolute values I pork meat usually and what are the recommendations for humans? Has the idea ration and/or absolute quantity achieved so far in pork using (any) feed ingredients / by any researchers? If not in pork, then in other meat sources? Why are you aiming to modify fatty acid composition in pork and not any other meat? Especially given that pork consumption is experiencing a long ongoing decline in many societies (probably exactly for its “unhealthy” reason)?

Omega-6 and omega-3 intake should be taken in a maximum ratio of 3: 1 which is the most optimal ratio for a pleasant 6/3 PUFAs. We also made attempts to influence poultry meat using fermented products, as poultry is the most consumed type of meat. Publications aimed at changing the ratio and composition of fatty acids, which we focused on. For the example:

Effect of fungal solid-state fermented product enriched with gamma-linolenic acid and ß-carotene on blood biochemistry and immunology of broiler chickens; Polish Journal of Veterinary Sciences 23(2):247-254, DOI: 10.24425/pjvs.2020.133639

Effect of Fungal Solid-State Fermented Product in Broiler Chicken Nutrition on Quality and Safety of Produced Breast Meat; September 2018, BioMed Research International 2018(5):1-8,  DOI: 10.1155/2018/2609548

Line 68: and what was the outcome of these research experiments?

Positive change of the n-6/n-3 PUFAs ratio

Line 68: I believe it was not the addition but rather the use of various oils. If you add oil you increase total energy of the diet, which brings many other problems. It is rather than feed formulation is altered to include “new” fat sources e.g. one fat source with a (more beneficial) different fatty acid composition replacing another. So please rephrase and do not use “addition” unless some feed ingredient was added “over the top” of a formulated diet.

We use “application” as a better suited word

Line 68: research has also used flaxseed oil in other species, evening primrose, sunflower oil, thistle oil, fish oils, fish meal. Please include, ideally in table form stating inclusion levels and outcomes.

Sentence was removed according to another reviewer. So it is not necessary to add the requested table.

Line 72-76: what was the inclusion level, duration of feeding and was the result statistically significant? These are only some relevant reasons why results vary so much. Without this context it does not become clear where the gap of knowledge is that your work is trying to fill and t also does not allow the reader to judge your results and critically evaluate your discussion.,

We added these information in to the material and methods section

Line 79: Please define “natural” antioxidant. Everything in this world comes from nature.

“….The possibility to incorporate natural vegetable based antioxidant in ….. “

Line 79: antioxidants are given in pig diets ever since pigs evolved. Vitamin E is an antioxidant and required for every pig, uptaken by wild pigs as much as commercial housed pigs. So its not a new approach at all, please be more precise as what you are trying to say.

We redefine the sentence to be more specific

Line 81: simultaneously administration of PUFA and what? PUFA alone will increase oxidation rate in meat.

n-3 PUFAs can decrease  aterogenic index of meat

Line 83: What specific substances are you referring to? What is your definition of “specific” substance?

We rewrite the sentence: “… oxidation of meat lipids and also enrich meat with selected fatty acids.”

Line 83: Please replace “supplemented” for reasons outlined above.

Sentence was removed according to another reviewer.

Line 84: The information in this sentence is not correct. Canola oil, soybean oil and linseed oil not only shorty chain fatty acids, but also essential fatty acids which are long chain, they also provide medium chain FA. I highly encourage the authors to critially revise the literatire on dietary fatty acids!

Sentence was removed according to another reviewer.

Line 86-90: Based on the information given until here it is not clear:

Why the specific concentration of flax seeds are used

We decided on the basis of studies available like Mathews et al. We wanted to observe a higher dose of a given flaxseed variety, with a high ALA content and its effect at a higher dose.

Why flaxseed was chosen to be investigated (is it a superior feed source over other dietary options, is it cheaper? More available? Etc)

In our conditions, flaxseed was more available compared to linseed oil. The physiological side of flaxseed addition and the effect on animal physiology were also monitored. Physiology data are not the subject of this publication

What changes in FA profile and ratio to expect (based on previous research, the fatty acid profile of flaxseed and the inclusion level in the diet, the expected concentration in the pig meat can be predicted), therefore making this study a proof of concept.

We expected a change in n-3 fatty acids. We wanted to prove the effect of shorter time and higher concentration of the supplement with the fact that the flax seed does not have a negative effect on the physiology of digestion (data not presented)

In what ways “intervals” (duration?) would be relevant or had been chosen to be investigated?

We chose them based on our experience with flaxseed

How antioxidants (line 79) play a role here- based on “simultaneous administration” explanation, why are antioxidants not investigated?

The study of antioxidants was not the subject of this study

Materials and Methods:

Line 94: please state the animal ethics approval number and authority.

Ethical commite of the university of veterinary medicine and pharmacy in Košice no. EKVP/2021-27.

How were the animals assigned to each group? Why was the flaxseed not taken into diet formulation but rather supplemented over the top? This would have impacted energy density and subsequently feed intake of the diet

We added requested information in to the manuscript

Table 1: please include detailed information about the diet composition listing all feed ingredients

Please take into consideration the nutrient dilution factor when adding the flaxseed over the top and provide calculated values of the final experimental diet that the pigs consumed  

Requested information are available at the lines 127-131.

Table 1 should also show the nutrient composition AS ANALYSED, since this always differs from the calculated composition.

Table 1 shows nutrient composition as analysed

Line 114: Please share the total nutrient and fatty acid profile of the flaxseed.

Nutrients was added into the table 1. We also add a new table of fatty acid profile of feed mixtures.

Table 1 also needs information about the microminerals, of special interest is the vitamin composition as some of them serve as antioxidants)

We did not investigate the microminerals.

Section 2.1: missing information about animal housing and killing: light, bedding, temperature, housing system, vaccination status, frequency of health checks, etc. The experiment is not reproducible with the current information given.

We added missing information into the manuscript part Mat and Mets (Line 109): ...Animals were kept in pens (groups of 2 animals per pen; 4.3 m2) equipped with nipple drinker. Average temperature was 18°C and humidity 60%. Health status, consistency of faeces and body weight of pigs were monitored at the begining, after 21 days of supplementation and at the end of experiment.  Animals underwent parasitological and bacteriological observations of faeces samples and nasal swabs with negative results.

Section 2.2 not clear how meat was processed before sample taking: animal transport and handling affects stress and therefore meat quality, stunning and killing method can affect meat quality as well as post- killing processing and time until sample removal. Please outline on all of it.

Pigs were housed at the Pig Fattening and Slaughter Station Inc. (Vajanského street 789, Spišské Vlachy, the Slovak republic), kindly see line 107 – 108, therefore stress during transportation was eliminated. Animals were slaughtered according to european standards (...možno tu doplniť nejaké číslo nariadenia...) and meat was imediately moved to chilling room with 4°C. So the meat quality was not affected.

Statistical Analysis:

What was your experimental unit, how many replicates, what was the statistical arrangement and design, how was data tested for normality, how treated if not normally distributed, how were outliers determined and handled.

We had 5 groups and each group consist from 6 animals. For meat samples we take sample from each animal and from the same anatomical part.

Analysis would have benefitted substantially from 3x2 factorial design taking 3 flaxseed inclusion levels (0, 5, 10%) and 2 feeding durations (3, 6 weeks) into account!!! Please change so you are able to investigate main effects and interaction.

We think the statistical analysis was chosen appropriately

Results:

Line 154: “In table 1 is shown…”

No comments

Line 158: replace “mixture” with “diet”

“ In Table 1 is shown the nutritional composition of pigs diet calculated … “

Line 159: negatively is a subjective opinion by the authors, please delete/replace with objective parameters (e.g. increased by xxx). Feeding too much or too little energy will have consequences of the pig owner’s finances to start with, so its all a matter or perspective.

“…. experimental diets, but the metabolizable energy balance was not influenced….”

All tables: Apply 3 – digit rule for better readability e.g. xx.x  x.xx

Tables 1,2,3 and 6 were rewritten according to the 3-digit rule

Discussion:

It is really hard to follow the discussion of the results without having the full information on the experimental diets available. Discussion needs to be more differentiated and critical as outlined in my comments on the introduction already. When referring to previous research results it should be clear in which species, at what inclusion level, fed for what duration. Especially since this manuscript covers dietary intake of pigs and humans there are very frequent unclarities, but also information is available on many other animal species which raises questions.  

The discussion was re-evaluated and rewritten on the basis of other reviewers. 

We believe that this revised manuscript is appropriate for publication.

Yours sincerely,

team of authors.

Reviewer 5 Report

It is a well written paper and solid study in general. Here are the main concerns:

  1. is there any nutrient values for flaxseed used in this study? 10% is a considerable number as an ingredient.
  2. what is the beginning and finishing weight of the pigs. If it is reported else where, please cited so readers can get access. Otherwise please list it as a result.
  3. Some reference in introduction are not correct. Please be more specify. 

Detailed comments are in the attachment.

Author Response

Dear reviewer, first of all thank you for your time, willingness and for your valuable comments, which helped us to improve the overall quality of the revised manuscript. All changes and modifications were highlighted with yellow color to indicate changes we have made in the revised text. We used professional English editing institution to improve the manuscript significantly.

is there any nutrient values for flaxseed used in this study? 10% is a considerable number as an ingredient.

Nutrient values of flaxseed used in this study were added to the table 1.

what is the beginning and finishing weight of the pigs. If it is reported elsewhere, please cited so readers can get access. Otherwise please list it as a result.

Beginning weight is mentioned in the line 99. Also the finishing weight will be added to the text, and as a new table as well.

Some reference in introduction are not correct. Please be more specify. 

References have been checked.

Line 24: The sentence could be revised to “Flaxseed with a concentration of alpha-linolenic acid (ALA) at 57% was used in the swine diet.”

Sentence was rewritten: Flaxseed with a conventration of alpha-linolenic acid (ALA) at 57% was used in the swine diet.

Line 25-26: The sentence could be revised to: Flaxseed supplementation at two levels (5 and 10%) in two time intervals (3 and 6 weeks before slaughter) was evaluated.

Sentence was rewritten:  Flaxseed supplementation at two levels (5 and 10%) in two time intervals (3 and 6 weeks before slaughter) was evaluated

Line 30 temperature.

Temperature value was added (+4°C)

Line 66-69: It’s better to combine two sentences into one and mention it as monogastric animals. The reference on second sentence is all about poultry species.

Sentences were rewritten and combined according to rewievers recommendations

Line 70: it should be “In monogastric animals, it is possible…”

Sentence was rewritten as: “In monogastric animals, it is possible to enhance the concentration of beneficial n-3 fatty acids in the muscle tissue since meat lipids reflect the nature of the dietary fat, feeding of the whole flaxseed to the pigs, increases the n-3 fatty acids in various tissues without the effect on growth and meat quality during storage. [8,9]”

Line 74-75: what is the effect of lose dose on swine?

We did not observe any dose loss effect

Line 76: “Juarez et al. [11] studied higher doses of flaxseed (up to 20%) without affecting the pigs’ growth performance.” Change to “Juarez et al. [11] reported higher doses of flaxseed supplementation up to 20% had no effect on growth performance”

Sentence was changed: “Juarez et al. [11] reported higher doses ….”

Line 77: “which is rich in a ALA” change to “which is rich in ALA”; I agree with your points, but I don’t think the reference 14 is related/supported your statement. Please double check

Sentence was changed: Flaxseed diet, which is rich in ALA, ….

Line 78: levels change to deposition.

Levels were changed for deposition

Line 80: I checked that systematic review, but there is no related statement on natural antioxidant and pork meat. Please label the source page of related statement instead of page 1-72 since this is a book.

Correct pages were added to the reference

Line 81: Simultaneously; administration of PUFAs rich ingredient

Sentence was rewritten according to reviewer

Line 83-85: The reference 14 is not relevant and sentence is not completed. Vegetable oil could only provide short chain n-3, but the flaxseed is the same thing.

Sentence was erased according to another reviewer´s request

Line 87: during two fattening periods

Sentence was rewritten according to reviewer

Line 88: on pork meat were investigated.

Sentence was rewritten according to reviewer

Line 97 “at” change to “in”

Part “at” was changed to “in”

Line 100: “with commercial complete feed mixture for pigs fattening” change to “a commercial complete fattening feed. What is the nutrient requirement of this commercial complete feed?

Sentence was changed to the commercial complete fattening feed. The nutritional requirements are in accordance with Slovak national legislation 440/2006 on General requirements for compound feeding stuffs

Line100-106: are those diet iso-caloric or iso-nitrous? Do you add the flaxseed on top of complete diet or re-formulate the diet?

We reformulate the diet to provide as much as possible same values compared to the control diet

Line 110-114: It’s better to summarize those info as a feed formulation table. Is the Table 1 calculated value or analyzed value. What is the nutrient matrix value for flaxseed (e.g. crude protein, ether extract, ME, avP, etc.).

Analysed flaxseed values were added. The values were analysed not calculated.

Line 134: of the loin, there is no “the”

Word “the” was erased

Line 150: how do you do multiple comparison? Tukey? Please justify. Table 2: It is better to reorder superscript a, b, c in the table. Either highest number labeled as a or c. This will make it easier for readers. For % of dry matter, I think there is a typo since 28.65 is a, 32.13 is b and 30.32 is c. Please add another row with p-values.

We use Tukey multiple comparison test. This was added to the statistical methods. Superscripts were reordered.

Line 170-172: is this description for m. gluteobiceps? Variation changes to content if it is.

Sentence was rewritten: The highest fat content was recorded in F10W3; and in F5W3 experimental group the highest protein content 21.95 ± 0.23 was observed in m.gluteobiceps

Line 174-175: add p value here since it is significant. Please justify which part of the tissue.

p-Value was added. Flaxseed supplementation for 6 weeks did not affect the dry matter and fat content for the m. longissimis dorsi samples (p > 0.05) from middle part of the muscle.

Line 175-176: for the m. longissimus dorsi samples; remove statistically. Remove statistically in later text and add p value.

p values were added

Line 187: remove statistical and put p value there instead.

p values were added

Line 189: Feeding flaxseed for 6 weeks had higher effect on lipid oxidation. Change to “Fats from pigs feeding flaxseed for 6 weeks are more susceptible to lipid oxidation during storage.”

Sentence was changed: Fats from pigs feeding flaxseed for 6 weeks are more susceptible to lipid oxidation during storage.

Line 244: in recent years

Sentence was changed: In recent years a healthy dietary ……

Line 259: Rephrase this sentence. Vitamin deficiency indeed affect desaturase and elongase, however, it is not relevant to the current trial.

Sentence was rephrased: Vitamin deficiency indeed affect desaturase and elongase, however, it is not relevant to the current trial.

Line 262: Sufficient income, but also the right ratios are essential factors for the organism. Please justify this sentence. I think you may mean a higher PUFAs and lower n-6/n-3 ratio will positively affect the fat deposition in pigs. Plus, the reference is about essential oils that is completely different from the topic of this research. Essential oils are plant extract from herbs and volatile generally.

Sentence was changed: Higher PUFAs and lower n-6/n-3 ratio will positively affect the fat deposition in pigs

Line 265: Phytochemicals and flaxseed oil are two different concepts. The first term is more like a GRAS herb medicine with no nutrient values, while flaxseed/flaxseed oil is more like a feed ingredient such as legume. Please modify your statement.

Sentence was erased according to another reviewer.

Line 266: Why does the fat sources and feeding intervals could change the fatty acids deposition? What makes monogastric animals special compared to ruminant in terms of fatty acids deposition?

Ruminants digest differently than monogastric animals. The processing of FA is different as the microbial flora in the rumen and the subsequent passage to the small intestine treats FA differently. Therefore, the direct addition of FA in monogastric animals is more easily affected.

Line 269: What is the effect of study from Kim et al?

Study of Kim et al show indication of the effect of linseed oil into diet (50g/kg). Kim observed that the addition of 50g of linseed oil can improve PUFA content n adipose tissue. But they also investigated the feed restriction effect on the finisher pigs, as they investigate also the regulation of ACC, FAS, LPS and HSL expression in adipose tissue.

Line 272: Please justify growth of PUFAs. Please connect the results from ref 23, 24, 25 to current study. Are the results similar?

Results was close to the our experiment. We consider them as important

Line 274: at the refrigeration temperature

Sentence was erased according to the another reviewer.

Line 276: Need reference on how oxidation affects sensory properties.

Sentence was erased according to another reviewer.

Line 278: Products including arachidonic acid (20:4n-6, ARA) and eicosapentaenoic acid (20:5n3, EPA) have various metabolic roles including eicosanoid production.

Sentence was rewritten: Products including arachidonic acid (20:4n-6, ARA) and …..

Line 280: what lipid supplementation? Tallow? Vegetable oil? Has shown an increase in the amount of….

Vegetable oils

Line 284: what is energy status? Do you mean ME levels or fat source or both? Please justify. Fat source is an obvious factor here. Poultry fat, tallow and vegetable fat are going to behave differently.

We mean metabolizable levels of fats

Line 298: how do you determine the ME here? By calculation from existing table or experiment Line 303: Are gilts or barrows used in this study?

            We calculate the ME from the experiment, We used gilts. The data was added to the manuscript.

Line 305: Please rephrase this sentence. It is confusing.

Sentence was erased according to another reviewer.

Line 314: length

Word lenght was rewritten to length

Line 322: Any justification on the difference between m. longissimus dorsi and m. gluteobiceps in LA and ARA content

            Fat storage in the removed muscle used for expriemet. M. gluteobiceps naturally has a higher proportion of intramuscular fat than m.longissimus dorsi this was also reflected in various LA values

Line 336: pork meat.

Port meat was changer to pork meat

Line 337: when palmitic and stearic acids are the dominant acids among the saturated fatty acids (SFAs). Please justify this statement.

Palmitic acid suggests that it is found, for example, in palm oil, but also in butter, cheese, milk and meat. Stearic acid is a fatty acid that occurs naturally in vegetable and animal fats

Line 348: is this change increase or decrease?

Sentence was erased according to another reviewer.

Line 378-383: you may want to move this paragraph to line 362 since it has nothing to do with TBARS here.

Paragraph was removed to the line 362

Line 400: administration of Vitamin E or other antioxidants indeed extends shelf life of lipid product. However, it is not really related to the current research. You can mention this as further research on how to protect the PUFAs in pork meat.

This part has been rewritten according to your recommendations

We believe that this revised manuscript is appropriate for publication.

Yours sincerely,

team of authors.

Round 2

Reviewer 2 Report

Thank you for your reply and taking my comments into account

Author Response

Dear reviewer, first of all thank you for your time, willingness and for your valuable comments, which helped us to improve the overall quality of the revised manuscript. We used professional English editing institution to improve the manuscript significantly.

We believe that this revised manuscript is appropriate for publication.

Yours sincerely,

team of authors.

Reviewer 4 Report

Line 102: I assume the heath status was checked daily and not only at the beginning and end of the experimental period as currently stated? 

Please amend table 1 (or provide a new table) by providing the feed ingredients (diet formulation) that resulted in the presented nutrient composition. This is crucial for other researcher to be able to repeat the experiment or interpret their data compared to yours as certain feed ingredients have dietetic value beyond the nutrients reported in table 1 (for example certain non-starch polysaccharides). 

Table 7 missing units (I assume kg?) , not sure why commas and not decimal points are used to report numbers? 

There are still minor formatting issues in the entire text which needs critical review with attention to detail. 

 All tables: please report p-values for the experimental factors and their interactions. 

Author Response

Dear reviewer, first of all thank you for your time, willingness and for your valuable comments, which helped us to improve the overall quality of the revised manuscript. All changes and modifications were highlighted with yellow color to indicate changes we have made in the revised text. We used professional English editing institution to improve the manuscript significantly.

Line 102: I assume the health status was checked daily and not only at the beginning and end of the experimental period as currently stated? 

Health status was monitored daily. We have supplemented the given information and edited the text to provide correct information. The animals were under the supervision of caregivers, doctors from the clinics of swine and doctors from the Department of Animal Physiology from the University of veterinary medicine and pharmacy in Košice ( line. 112)

Please amend table 1 (or provide a new table) by providing the feed ingredients (diet formulation) that resulted in the presented nutrient composition. This is crucial for other researcher to be able to repeat the experiment or interpret their data compared to yours as certain feed ingredients have dietetic value beyond the nutrients reported in table 1 (for example certain non-starch polysaccharides). 

Information of the feed composition and amounts of each feed ingredients are provided in lines 127-131.

Table 7 missing units (I assume kg?) , not sure why commas and not decimal points are used to report numbers? 

Units were added to the Table. Thank you for your comment and correction. Unfortunately we forget to switch our table editor to English during preparation of the table, this was the problem with the commas. Decimal points were added.

There are still minor formatting issues in the entire text which needs critical review with attention to detail. 

Formatting was checked.

 All tables: please report p-values for the experimental factors and their interactions. 

p-Values were added to the tables.

We believe that this revised manuscript is appropriate for publication.

Yours sincerely,

team of authors.
